# Differential Expression of miRNAs, lncRNAs, and circRNAs between Ovaries and Testes in Common Carp (*Cyprinus carpio*)

**DOI:** 10.3390/cells12222631

**Published:** 2023-11-15

**Authors:** Mingxi Hou, Qi Wang, Jin Zhang, Ran Zhao, Yiming Cao, Shuangting Yu, Kaikuo Wang, Yingjie Chen, Ziyao Ma, Xiaoqing Sun, Yan Zhang, Jiongtang Li

**Affiliations:** 1Key Laboratory of Aquatic Genomics, Ministry of Agriculture and Rural Affairs and Beijing Key Laboratory of Fishery Biotechnology, Chinese Academy of Fishery Sciences, Beijing 100141, China; houmingxi@cafs.ac.cn (M.H.); wangqi@cafs.ac.cn (Q.W.); zhangjin@cafs.ac.cn (J.Z.); zhaoran@cafs.ac.cn (R.Z.); caoyiming@cafs.ac.cn (Y.C.); styuwork@163.com (S.Y.); sunxiaoqing@cafs.ac.cn (X.S.); zhangyan@cafs.ac.cn (Y.Z.); 2Chinese Academy of Agricultural Sciences, Beijing 100081, China; 3National Demonstration Center for Experimental Fisheries Science Education, Shanghai Ocean University, Shanghai 201306, China; 18631836881@163.com (K.W.); cyjttkl@163.com (Y.C.); zql3703700@163.com (Z.M.)

**Keywords:** non-coding RNAs, sex differentiation, common carp, expression profiles, competing endogenous RNAs

## Abstract

Female common carp grow faster than male individuals, implying that rearing females could be more profitable in aquaculture. Non-coding RNAs (ncRNAs) serve as versatile regulators with multiple functions in diverse biological processes. However, the roles of ncRNAs in the sex differentiation of common carp are less studied. In this study, we investigated the expression profiles of ncRNAs, including miRNAs, lncRNAs, and circRNAs, in the gonads to comprehend the roles of ncRNAs in sex differentiation in common carp. A substantial number of differentially expressed (DE) ncRNAs in ovaries and testes were identified. Some miRNAs, notably miR-205, miR-214, and miR-460-5p, might modulate hormone synthesis and thus maintain sex. A novel miRNA, novel_158, was predicted to suppress the expression of *foxl3*. DE lncRNAs were associated with oocyte meiosis, GnRH signaling pathways, and steroid biosynthesis, while DE circRNA target genes were enriched in the WNT signaling pathway and MAPK signaling pathway. We also analyzed ncRNA-mRNA interactions to shed light on the crosstalk between competing endogenous RNAs (ceRNAs), which is the critical mechanism by which lncRNAs and circRNAs function. Some lncRNAs and circRNAs may be able to competitively bind novel_313, a new miRNA, and thus regulate *hsd17β3*. Our research will provide a valuable resource for understanding the genetic basis of gonadal differentiation and development in common carp.

## 1. Introduction

Sex determination and sex differentiation are a classic and challenging topic that has fascinated researchers for generations [1]. Given that sexual dimorphism in growth, size, color, and other economic characteristics is prevalent, sex is also one of the most valuable traits in aquaculture animals [2,3]. For fish with sexual dimorphism in growth, sex-controlled breeding to produce monosexual populations can increase economic effectiveness [4]. Therefore, research shedding light on the molecular mechanisms of sex differentiation in teleosts has the potential to bring enormous value to the aquaculture industry.

Non-coding RNAs (ncRNAs), a class of transcribed RNA molecules that do not encode proteins, perform crucial roles in gene expression regulation and various biological processes [5]. Small RNAs (smaRNAs), long non-coding RNAs (lncRNAs), and circular RNAs (circRNAs) are the most common types of ncRNAs. The smaRNAs can be classified into several categories, including microRNAs (miRNAs), small interfering RNAs (siRNAs), piwi interacting RNAs (piRNAs), and others [6]. Of these, microRNAs (miRNAs) with a size of 21–25 nt are the most known and have been extensively studied, and they negatively regulate gene expression by pairing with the corresponding target mRNA sequences [7]. They play crucial roles in gonadal development and gametogenesis by inhibiting classically genes related to sex differentiation. For example, miR-124 inhibits the expression of *sox9* in the ovaries [8], and miR-107, highly expressed in the testes, mediates the estrogen signaling pathway by directly targeting *cyp19a1a* [9]. The lncRNAs are long ncRNAs with a length over 200 nt [10]. The lncRNA TESCO interacts with the transcription factor SF1, which sustains the expression of *Sox9* and testicular development [11]. The lncRNA DMR plays a role in sexual differentiation by negatively regulating the expression of *Dmrt1* [12]. The circRNAs are characterized by a covalently closed-loop structure without terminal 5′ caps and 3′ polyadenylated tails [13,14]. They can act as miRNA sponges to regulate gene expression by preventing them from binding to their target mRNAs [15]. However, there are only a few studies on the role of circRNAs in sex differentiation of the teleost.

The common carp (*Cyprinus carpio*) is a XX/XY sex-determination system Cyprinid fish [16]. It is widely distributed worldwide and has become one of the most critical commercially cultivated freshwater fish species due to its adaptation to complex and changing environments and rapid growth rate [17]. The global production of common carp was about 4.41 million metric tons, accounting for roughly 5.2% of the total freshwater fishery production in 2021 [18]. More importantly, the common carp shows apparent sexual dimorphism in growth, with females growing faster than males [19]. Studying the molecular mechanisms of sexual differentiation will help conduct sex-controlled breeding. The chromosome-level whole-genome sequence of common carp is available [20,21]. Although many studies have been performed to analyze the differential expression (DE) genes between the testes and ovaries [18,22,23], the function of ncRNAs in sexual differentiation has not been reported. There is room for improvement in functional genomics research in common carp, and more omics data are required.

In the present study, we analyzed the expression profiles of ncRNAs in the ovaries and testes. We also analyzed ncRNA-mRNA interactions to investigate the potential role of competing endogenous RNA (ceRNA) crosstalk in sexual differentiation and maintenance. Our findings will contribute to the understanding of the genetic basis of sex differentiation in common carp.

## 2. Materials and Methods

### 2.1. Ethics Statement

All experiments complied with the principles of animal care and use for scientific purposes established by the Animal Care and Use Committee of the Chinese Academy of Fishery Sciences (Protocol code ACUC-CAFS-20220615). In addition, the animals were sedated with MS222 (40 mg/L) to minimize discomfort before surgery.

### 2.2. Sample Collection and RNA Extraction

The common carp used in this study were reared at the Fangshan Experimental Station in Beijing, China. Ovary and testis tissues, with three replications in each group, were dissected from sexually mature carp and then frozen in liquid nitrogen. All samples were stored in a −80 °C refrigerator for later use. Total RNA was extracted from ovaries and testes using TRIzol reagent (Invitrogen, Carlsbad, CA, USA). Briefly, the sample was fully homogenized in TRIzol and incubated at room temperature for 5 min. Next, we added chloroform, shook the mix thoroughly, incubated it for 3 min, and centrifuged it for 15 min at 12,000× *g* at 4 °C. The aqueous phase was transferred to a new tube and treated with DNase I to digest the residual DNA. Total RNA precipitate formed a white pellet after adding isopropanol and centrifugation. The concentration of total RNA was determined by using Nanodrop One spectrophotometer (Thermo Fisher, Waltham, MA, USA), and the quality was checked by using Agilent Bioanalyzer 2100 system.

### 2.3. Library Preparation and Sequencing

High-quality RNAs with the NanoDrop 260/280 ratios of approximately 2.0, RNA Integrity Number (RIN) ≥7.5 for lncRNA and circRNA, and RIN ≥8.5 for smaRNA were used for further library construction. After removing ribosomal RNAs, lncRNA libraries were generated using NEBNext^®^ UltraTM RNA Library Prep Kit for Illumina^®^ (NEB, Newton, MA, USA). Before constructing the circRNA libraries, the linear RNAs needed to be digested with RNase R (Epicentre, Verona, WI, USA), and the circRNA libraries were generated with the NEBNext^®^ UltraTM Directional RNA Library Prep Kit for Illumina^®^ (NEB). For smaRNA libraries, the 3′ and 5′ adaptors were ligated to the 3′ and 5′ ends of the smaRNAs, respectively, and then smaRNA libraries were constructed using NEB Next^®^ Multiplex SmaRNA Library Prep Set for Illumina^®^. The Agilent 2100 Bioanalyzer system was used to assess the quality of libraries. Then, lncRNA and circRNA libraries were sequenced on the Illumina NovaSeq 6000 platform to produce 150 bp paired-end reads. The smaRNA libraries were sequenced on the Illumina Hiseq 2500 platform (Novogene, Beijing, China) to produce 50 bp single-end reads. 

### 2.4. Data Quality Control and Read Alignment

Clean data were obtained by removing reads containing adapters, poly-N, and low-quality reads from the raw data. The common carp reference genome was downloaded from NCBI [21]. Paired-end clean reads generated from lncRNA and circRNA sequencing were aligned to the reference genome using Hisat2 (v2.0.5). For smaRNA analysis, clean reads of the length between 18 and 35 nt were selected for mapping the reference genome using Bowtie (v2.4.0) [24].

### 2.5. Bioinformatics Analysis of smaRNA Sequences

The reads mapped to the reference genome were aligned with the miRNA sequences in miRbase (v22.1) [25] to identify known miRNAs. The miREvo [26] and mir-deep2 [27] were used to predict novel miRNA based on the secondary hairpin structure of miRNA precursors. The target gene of miRNA was predicted using miRanda. The miRNA editing analysis was conducted using the MiRME method [28]. For miRNA family analysis, miFam.dat (http://www.mirbase.org/ (accessed on 1 December 2022)) was used for known miRNA, while Rfam (http://rfam.sanger.ac.uk/search/ (accessed on 3 December 2022)) was used for predicting novel miRNA.

### 2.6. Processing of lncRNA Sequencing Data

StringTie (v1.3.1) was used to reassemble and quantify transcriptome based on the reads mapping to the reference genome to obtain better accuracy and integrity transcripts [29]. Three coding potential analysis software programs, including Coding-Non-Coding Index (CNCI) (v2.0) [30], Coding Potential Calculator (CPC) (v3.2.0) [31], and Pfam ScanC (PFAM) (v1.6) [32], were combined to screen non-protein-coding RNA candidates. The candidates that were not annotated in NCBI were predicted as lncRNAs. Target genes of lncRNAs were predicted by co-location and co-expression analysis between lncRNAs and protein-coding genes, and the threshold for co-location analysis was set as 100 kb upstream or downstream of the lncRNAs [33,34]. rMATS (v4.1.0) software was used for AS event analysis with default parameters.

### 2.7. Identification of circRNAs and Prediction of Coding Potential

The circRNAs were identified using find_circ (v1.0) [35] and CIRI (v2.0.5) [36] using reads mapping to the reference genome. The circRNAs with internal ribosome entry sites (IRES) have the potential to be translated into proteins. The IRES elements of circRNAs were first predicted using the IRESfinder software (v1.0), and then CNCI, CPC, and PFAM were combined to validate the coding potential of the circRNA sequences [37].

### 2.8. Differential Expression Analysis

Fragments Per Kilobase of transcript sequence per Millions of base pairs sequenced (FPKM) for lncRNAs and mRNAs and transcripts per millions (TPM) for miRNA and circRNA were used to normalize the expression levels. Differential expression analysis was performed using the edgeR R package (v3.22.5) based on FPKM values and DESeq2 R package (v1.20.0) based on TPM values [38,39]. Corrected *p*-values ≤ 0.05 were set as the threshold to mine expressed transcripts.

### 2.9. Enrichment Analysis of Differential Candidate Genes

The differential candidate genes were submitted for functional enrichment analysis using the ClusterProfiler R package (v3.9), which included Gene Ontology (GO) and Kyoto Encyclopedia of Genes and Genomes (KEGG) pathway terms [40]. *p*-Values lower than 0.05 are considered significant, and the top terms were selected for visualization.

### 2.10. Quantitative PCR

The expression level of mRNA was detected by stem-loop quantitative PCR [41]. The complementary DNA (cDNA) for miRNA real-time PCR was reverse-transcribed using the GoScript™ Reverse Transcription System (Promega, Madison, WI, USA), and the cDNA for mRNA, lncRNA, and circRNA were synthesized using the Rever Tra-Ace M-MLV kit (TOYOBO, Japan). The real-time quantitative PCR was conducted on the Bio-Rad CFX96 Touch Detection System (Bio-Rad, Hercules, CA, USA). Each reaction system consisted of 10 μL of 2× SYBR Green mix (TOYOBO, Tokyo, Japan), 7 μL of ddH2O, 0.5 μL of forward and reverse primers, and 2 μL of cDNA. The reaction program was as follows: 95 °C for 3 min, followed by 40 cycles of 95 °C for 15 s, 60 °C for 15 s, and 72 °C for 15 s. For miRNA quantitative PCR, U6 was used as the reference gene, while 40 s was used as the reference gene for mRNA, lncRNA, and circRNA. The relative expression level was calculated using the 2^−ΔΔCt^ method. Student *t*-test was used to analyze statistical significance. *p* ≤ 0.05 indicated a statistically significant difference.

### 2.11. ceRNA Analysis

The competing endogenous RNA (ceRNA) hypothesis, a core mechanism for the action of ncRNAs, suggests that lncRNAs or circRNAs can interact with miRNAs and keep them from binding with mRNAs. The miRanda software (v3.3a) [42] was used to predict the interactions of mRNAs-miRNAs, lncRNAs-miRNAs, and circRNAs-miRNAs. lncRNA-mRNA or circRNA-mRNA pairs with the same miRNA binding site were calculated (*p*-value ≤ 0.05). The ceRNA networks were visualized using Cytoscape (v3.10.0).

## 3. Results

### 3.1. Overview of Gonadal smaRNA Sequencing of Common Carp

The ovaries and testes of common carp were sampled and sequenced in the present study. smaRNA sequencing revealed a total of 70,471,545 raw reads from common carp gonads. After quality filtering, 34,805,668 clean readings were obtained from the ovaries and 34,658,888 from the testes. Reads ranging in length from 18 to 35 nt, making up 95.57% of all reads, were picked out for further analysis (Appendix A). In both ovaries and testes, the most prevalent size of smaRNA is 27–28 nt, followed by 26 and 29 nt (Figure 1A). With 92.48% of filtered reads mapping the reference genome, smaRNA sequences were heavily mapped to chromosomes NC_056574.1, NC_056575.1, and NC_056599.1. The distribution density of reads on the top 10 chromosomes is shown in Figure 1B. Housekeeping ncRNAs such as ribosomal RNA (rRNA), transfer RNA (tRNA), small nuclear RNA (snRNA), and small nucleolar RNA (snoRNA) were filtered out and counted, as shown in Appendix A.

### 3.2. Analysis of miRNA in Common Carp Gonads

In the present study, 2,765,443 reads were mapped to 141 known mature miRNAs by comparison with the miRBase database, and 57 novel mature miRNAs were identified by predicting the hairpin structure of their precursor sequences using the remaining reads without annotated information. The details of the miRNA sequence are shown in Appendix A. The first base of a mature miRNA sequence is highly biased due to the cleavage of the mature miRNA from the precursor miRNA by the enzyme Dicer. A considerable proportion of known and novel miRNAs showed a base bias at the first position toward U, suggesting that the miRNAs identified in this study are reliable (Figure 1C). The family analysis was performed on the identified miRNAs to explore conservation and presence in other species (Appendix A). The target genes of miRNAs may be different due to one or two base changes in the seed region [43], and the result of miRNA base editing analysis is shown in Appendix A.

### 3.3. Differential Expression of miRNAs between Ovaries and Testes

We normalized the expression levels with TPM to detect the DE miRNAs between ovaries and testes, as shown in Figure 2A. A high correlation between the different samples indicates that the data met the requirements of the subsequent analysis (Figure 2B). Among all the miRNAs identified, 11 are only present in the ovaries, while 11 are only in the testes (Figure 2C). We identified 63 miRNAs up-regulated and 61 miRNAs down-regulated in the ovaries compared to the testes (Figure 2D; Appendix A). Of all the DE miRNAs, there are 91 known and 33 newly identified miRNAs, including miR-125b and miR-153b, which are known to be involved in sex differentiation. The 124 DE miRNAs were grouped into six subclusters with various expression patterns. Subclusters 1, 4, and 6 were down-regulated in testes, while subclusters 2, 3, and 5 were up-regulated in testes (Figure 2E; Appendix A).

### 3.4. Target Gene Prediction and Functional Enrichment of miRNAs

The miRNAs identified in this study were predicted to target 6776 genes (Appendix A). miR-205, miR-214, and miR-460-5p may be involved in inhibiting estrogen receptors, novel_158 and novel_313 may target *foxl3* and *hsd17β3*, respectively. The results of KEGG analysis showed that 73 terms were enriched, and the top 20 terms were shown in Figure 2F. Among these, the TGF-beta signaling pathway is a classical pathway involved in sex differentiation. GO analysis showed that the target genes of the DE miRNAs were involved in G-protein coupled receptor activity and membrane components and were heavily involved in hormone-related pathways such as hormone activity and cellular response to hormone stimuli (Figure 2G).

### 3.5. Overview of Gonadal lncRNA Sequencing

The lncRNA sequencing yielded 38.18 G raw data from the ovaries and 40.39 G from the testes. After removing the low-quality reads, we retained 36.91 G clean reads for the ovaries and 39.31 G for the testes. A total of 87.80% of clean reads in the ovary libraries and 83.38% in testis libraries were then efficiently mapped to the common carp reference genome (Appendix A). We identified 13,911 lncRNA transcripts, including 7684 known and 6227 newly discovered (Appendix A). A total of 56.7% of the newly discovered lncRNAs were mapped in the intergenic regions (Figure 3A). mRNA and lncRNA transcript characteristics, including length, exon number, and open reading frame length, suggest that our predictions are highly accurate and reliable (Figure 3B). In addition, 252 novel mRNAs were identified in this study (Appendix A).

### 3.6. Differential Expression of lncRNAs

The FPKM box plot (Appendix A) shows the overall expression of transcripts in each sample. To ensure the reliability of the data, we conducted a correlation analysis (Appendix A) and PCA analysis (Appendix A). A total of 7863 lncRNA transcripts were differentially expressed. Of DE lncRNAs, 1764 transcripts were up-regulated, while 6099 were down-regulated in ovaries compared to testes (Figure 4A,B; Appendix A). The cluster analysis demonstrates consistent expressions within groups and differential expressions in different groups (Figure 4C). To identify the target genes of lncRNAs, we performed co-expression and co-location analysis of lncRNAs and protein-coding genes (Appendix A). Target genes *smad1* and *esr1* were identified by co-expression analysis and *zp3* by co-location analysis. Differentially expressed target genes of lncRNAs were then subjected to GO and KEGG enrichment analysis to investigate the potential function. For target genes predicted from co-expression, GO analysis revealed enrichment of transcription factor complexes, DNA conformation changes, and pathways related to the cytoskeleton (Figure 4D), while the GnRH signaling pathway and oocyte meiosis were enriched in the KEGG enrichment (Figure 4E). The KEGG enrichment of target genes predicted from co-location analysis showed that the steroid biosynthesis was enriched (Appendix A).

### 3.7. Differentially Expressed mRNAs

We also identified 34,405 sex-biased expression mRNAs in this study, including 15,653 highly expressed in the ovaries and 18,752 highly expressed in the testes (Appendix A). The well-known genes *dmrt1*, *amh*, and *gsdf* were up-regulated in the testes, and the female differentiation star genes *foxl2* and *cyp19a1a* and the ZP family genes were expressed highly in the ovaries (Appendix A). The common carp is an allo-tetraploid species, and we also analyzed the subgenomes to which the DE mRNA belonged. There were 15,930 and 17,236 DE mRNAs belonging to subgenomes A and B, respectively. The remaining DE mRNAs were transcribed from mitochondria or unplaced genomic scaffolds (Appendix A). GO analysis revealed that differentially expressed genes are involved in DNA metabolism, the regulation of the cell cycle, and transcription factor activity (Appendix A). KEGG enrichment showed that DE genes are enriched mainly in the oocyte meiosis, cell cycle, and ErbB signaling pathways (Appendix A). The ErbB signaling pathway can activate PI3K/AKT and MAPK signaling pathways to modulate gonadal differentiation.

Alternative splicing (AS), which can generate multiple transcripts from the same precursor mRNA, regulates various physiological processes, including testicular and ovarian development [44]. We analyzed five kinds of alternative splicing events: skipping exon, intron retention, alternative 3′ splice site, alternative 5′ splice site, and mutually exclusive exons. We identified 24,976 AS events. Skipping exon is the major AS event, accounting for 69.23%. Moreover, 3893 differential AS events exist between ovaries and testes (Appendix A). A case of the AS pattern of *akap13* is shown in Appendix A.

### 3.8. Identification of Gonadal circRNAs

In total, we obtained 55.93 G circRNA sequencing raw data, and after data trimming by removing low-quality reads, ~344 million clean reads were retained. On average, 93.53% of clean reads in the ovaries and 85.40% in the testes were mapped to the reference genome (Appendix A). Then, 6611 circRNAs were identified using CIRI and find_circ software, and 6603 RNAs were retained for subsequent analysis after removing the duplicates (Appendix A). A total of 90.29% of circRNAs were transcribed from exon regions, while only 4.63% and 5.08% of circRNAs were transcribed from intron and intergenic regions, respectively (Figure 5A). The average length of circRNAs in the ovaries was approximately 284 nt, while the average length in the testes was approximately 298 nt. More than 60% of circRNAs were between 200 nt and 400 nt in length, and only 3.37% were longer than 500 nt (Figure 5B).

### 3.9. Expression Profiles and Functional Analysis of circRNAs in Gonads

The expressions of circRNAs were adjusted using TPM, and the TPM box plots showed the expression profiles of each sample (Figure 6A). According to the volcano map, 2082 circRNAs showed differential expression between ovaries and testes, of which 1047 circRNAs were highly expressed in ovaries while 1035 were highly expressed in testes (Figure 6B; Appendix A). Clustering analysis revealed that the expression patterns of circRNAs were consistent within each group and distinguishable between ovaries and testes (Figure 6C). We analyzed the function of genes transcribing DE circRNAs. GO enrichment revealed that they were associated with calcium channel activity, cytoskeleton protein binding, and transition metal ion binding (Figure 6D). KEGG analysis revealed enriched pathways involved in oocyte meiosis, WNT signaling, MAPK signaling, and calcium signaling (Figure 6E). Furthermore, certain circRNAs possess coding capacity, and we conducted a circRNA coding potential prediction, as presented in Appendix A.

### 3.10. Validation of Expression Profiles by qPCR

The relative expression levels of four miRNAs, lncRNAs, mRNAs, and circRNAs were examined by qPCR, respectively. Primers used for qPCR are listed in Appendix A. As shown in Figure 7, miR-122, miR-192, XR_002016692, XR_002020130, *cyp19a1a*, *zp3*, novel_circ_0006872, and novel_circ_0010650 showed ovary-biased expression, while miR-153b, miR-93, XR_006158258, XR_006156779, *dmrt1*, *sox9a*, novel_circ_0003495, and novel_circ_0011083 showed testis-biased expression. The expression trends of the 16 selected RNAs by qPCR were consistent with those of the transcriptomic profile analysis, demonstrating the reliability of the sequencing data.

### 3.11. Complex Interactions among mRNAs and ncRNAs

The crosstalk between the competing endogenous RNAs (ceRNAs) is a vital mechanism by which ncRNAs function. lncRNAs or circRNAs can sponge off miRNAs and then regulate gene expression. We analyzed the binding sites of miRNAs on lncRNAs and circRNAs and then established the lncRNA/circRNA-miRNA-mRNA regulatory relationship (Appendix A). For example, 17-beta hydroxysteroid dehydrogenase 3, which catalyzes the conversion of androstenedione to testosterone, is encoded by the *hsd17β3* gene. We predicted that multiple circRNAs and lncRNAs might compete with *hsd17β3* mRNA for binding to a new miRNA named novel_313. These circRNAs and lncRNAs might act as spongers to regulate the expression of *hsd17β3* (Figure 8; Appendix A).

## 4. Discussion

In recent years, a growing body of research has suggested that RNA molecules exhibit more functions than simply serving as templates for protein synthesis [5]. The proportion of ncRNAs in all transcripts is much higher than mRNAs, and ncRNAs can interact with DNA, proteins, and other RNA molecules to orchestrate most biological processes [45,46]. The function of ncRNAs in regulating sex differentiation in teleosts requires further investigation. In the present study, we systematically analyzed the expression profiles of ncRNAs between ovaries and testes, revealing numerous sex-specific ncRNAs in common carp.

Many studies have shown that lncRNAs can regulate well-known genes or signaling pathways to perform biological functions [47,48]. We identified 13,911 lncRNA transcripts, of which 6227 were novel in the gonads of common carp. Comparison of ovarian and testicular differences revealed differential expression of 7863 lncRNAs whose target genes are associated with vital biological processes, including steroid biosynthesis, oocyte meiosis, and GnRH signaling pathways. It suggested that lncRNAs may play a crucial role in gonadal development by regulating hormone synthesis. Although sex determination genes vary considerably among different fish species, the downstream genes and pathways of sex differentiation tend to be more conserved [49,50]. We analyzed the differential expression of mRNA. The results showed that the well-studied genes *foxl2* and *cyp19a1a* were highly expressed in the ovaries, while *dmrt1* and *amh* were highly expressed in the testes, indicating the reliability of our results. Previous studies show that genome duplication can increase the complexity of gene regulatory systems and provide an opportunity for the gain or loss of gene function [51]. Our results showed 15,930 mRNAs transcribed from the A subgenome and 17,236 transcribed from the B subgenome, indicating that both A and B subgenomes play vital roles in regulating sex differentiation.

Most circRNAs contain multiple miRNA binding sites, allowing them to bind and sequester miRNAs, thereby preventing miRNAs from interacting with their target mRNAs [35]. In addition, some circRNAs can undergo translation and produce functional proteins [52]. In mice, a testis-specific circRNA, circSry, regulates the functional implementation of the sex-determination gene Sry by competitively binding to miRNA-138 [15]. We identified 6603 circRNAs in the gonads of common carp, of which 2064 showed differential expression between ovaries and testes. Enrichment analysis of DE circRNAs indicates that WNT signaling and MAPK signaling pathways were enriched. These pathways are known to play pivotal roles in preserving ovarian differentiation and regulating the proliferation of primordial germ cells, respectively [53,54]. circRNAs may potentially contribute to sex differentiation in common carp through the involvement of the signaling mentioned above pathways. In addition, we analyzed the coding potential of circRNAs in this study.

The miRNAs are some of the most widely studied smaRNAs, and they are conserved across different species [55,56]. The analysis of the length distribution of smaRNAs revealed that the size range 26–29 nt is the most prevalent size. This finding may be attributed to the higher expression of piRNAs, a class of smaRNAs with lengths of 25–30 nt, which are mainly expressed in the germline and play a crucial role in germline development [57]. Our research is consistent with the findings of previous studies conducted on the zebrafish (*Danio rerio*) and tilapia (*Oreochromis niloticus*) [58,59].

Mature miRNAs can modulate the expression of target genes by recognizing and binding to homologous mRNA sequences. Previous studies have demonstrated that miR-17-5p and miR-20a may inhibit the expression of *dmrt1*, while miR-138, miR-200a, and miR-338 may negatively regulate the expression of *cyp17a2* in tilapias [60]. In the orange-spotted grouper (*Epinephelus coioides*), miR-26a-5p may act as a regulatory factor of *cyp19a1a* to modulate the production of estradiol-17β (E2), the primary functional estrogen [61]. miR-153b-3p, predominantly expressed in the testes of common carp, functions to fine-tune the complex process of male germ cell proliferation and differentiation by directly targeting *amh* [62]. In the current study, we identified 198 mature miRNAs, of which 57 are novel. There are 124 differentially expressed miRNAs, including miR-112, miR-125b, and miR-153b. The prediction of the miRNA target gene suggests that miR-205, miR-214, and miR-460-5p may participate in inhibiting several estrogen receptors. *foxl3*, which is related to influencing the sex fate decision in the medaka (*Oryzias latipes*) and tilapia [63,64], may be regulated by novel_158 in the common carp. Additionally, the differentially expressed miRNAs have a function in the TGF-beta signaling pathway, a traditional pathway that regulates sex differentiation and hormone-related pathways. These findings imply that miRNA plays a critical role in sex differentiation and maintenance in common carp.

The ceRNA crosstalk is recognized as a critical mechanism for ncRNA action. Many lncRNAs or circRNAs are known as miRNA sponges since they contain numerous high-affinity miRNA binding sites. They can bind and sequester particular miRNAs, preventing them from interacting with target mRNAs [65]. For example, a circRNA called circdmrt1 and a lncRNA called AMSDT were up-regulated in the Chinese tongue sole (*Cynoglossus semilaevis*) testes. These two ncRNAs can bind to cse-miR-196 and hence abolish its endogenous suppressive effect on the target gene *gsdf* [66]. In the current study, we analyzed ncRNA-mRNA interactions in common carp gonads to illuminate potential ceRNA crosstalk in sex determination and differentiation. We predicted that several circRNAs and lncRNAs might compete with *hsd17β3* mRNA to regulate its expression.

## 5. Conclusions

We yielded a substantial pool of ncRNAs associated with sex differentiation in the present study. This valuable collection of scientific data will serve as a valuable resource not only for understanding the underlying genetic basis of gonad differentiation and development but also for sex-control breeding in common carp.

## Figures and Tables

**Figure 1 cells-12-02631-f001:**
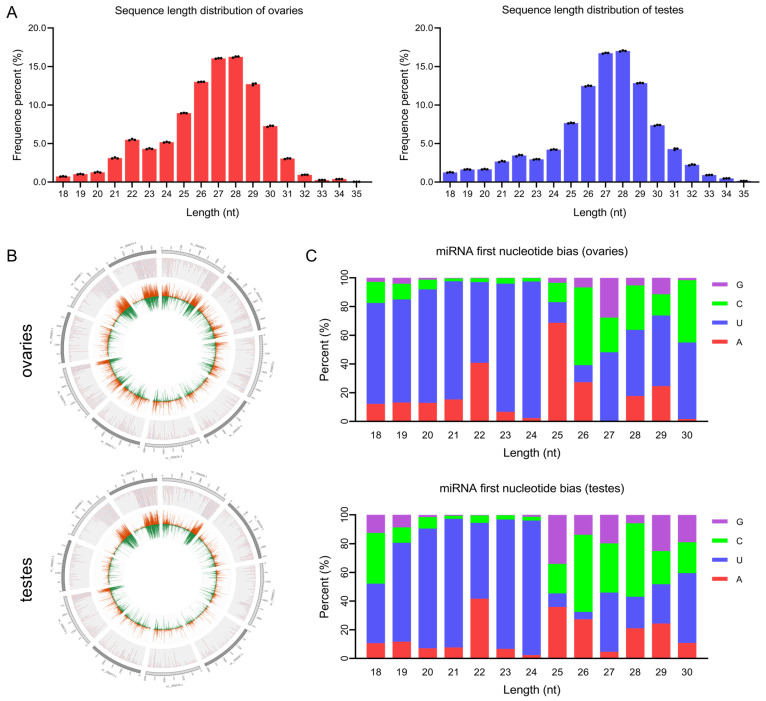
Characterization of gonadal smaRNAs and miRNAs in common carp. (**A**) Length distribution of smaRNA sequences in the ovaries and testes. (**B**) Distribution of gonadal miRNAs on chromosomes. The outermost circle is designated as the chromosome to be displayed. Orange and green represent reads mapped to positive and negative chains, respectively. (**C**) First base preference of known miRNAs ranging in length from 18 to 30 nt in the ovaries and testes. The horizontal axis represents the length of the miRNA, while the vertical axis represents the ratio of A/U/C/G in the first base.

**Figure 2 cells-12-02631-f002:**
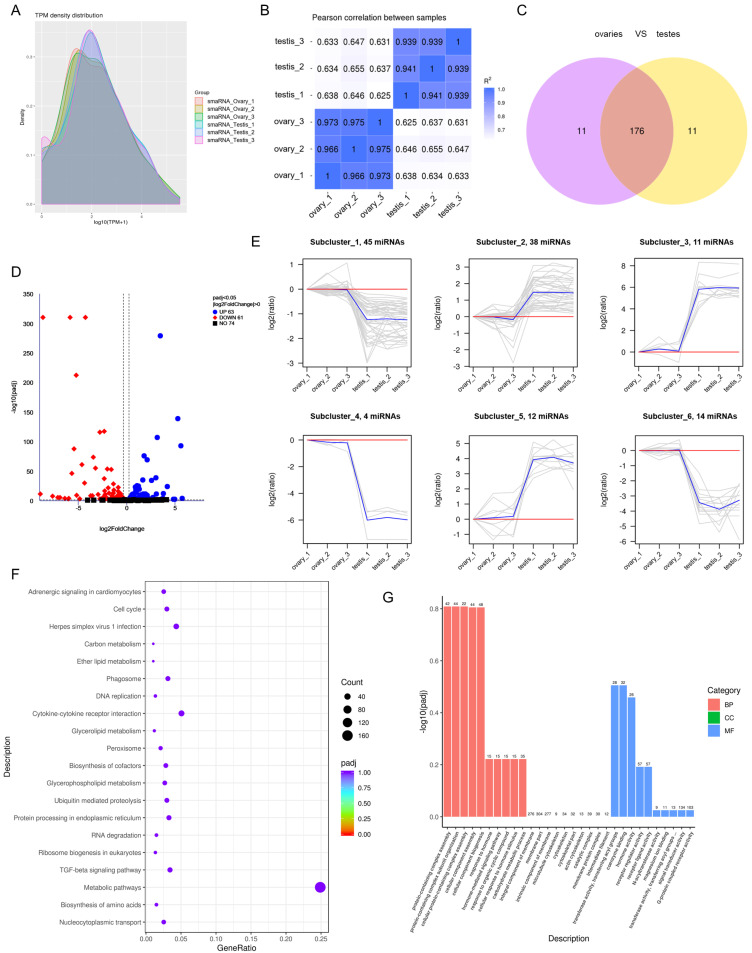
Identification of differentially expressed miRNAs and functional analysis of their target genes in ovaries and testes. (**A**) TPM density distribution of different samples. The horizontal axis is the log10 (TPM + 1) value, and the vertical axis is the density distribution. (**B**) The correlation analysis of different samples. (**C**) Venn diagram of miRNAs expressed in ovaries and testes. (**D**) Volcano map of differentially expressed miRNAs. Blue dots represent the up-regulated miRNAs, and red dots represent the down-regulated miRNAs in ovaries compared to testes. (**E**) Clustering of the 124 DE miRNAs into six subclusters. Gray lines represent the relative expression levels of each gene; blue lines represent the mean value in each subcluster; red lines represent the reference. (**F**) The KEGG pathways with the lowest *p*-value by DE miRNA target genes. (**G**) The GO terms with the lowest *p*-value by DE miRNA target genes. BP: biological process; CC: cell component; MF: molecular function.

**Figure 3 cells-12-02631-f003:**
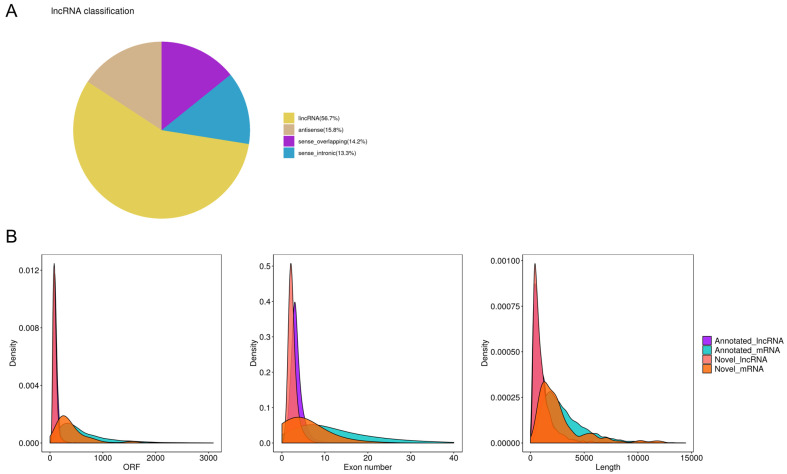
Characteristics of novel lncRNA transcripts identified in this study. (**A**) Classification and proportion of novel lncRNAs. antisense: overlapped with the antisense strand of one or more exons of a protein-coding gene; lincRNA: located in the intergenic region; sense overlapping: overlapped with sense strand of one or more exons of a protein-coding gene; sense intronic: located in the intronic region of a coding gene and does not intersect with any of exons on the same strand. (**B**) Comparison of the characteristics of all the lncRNAs and mRNAs.

**Figure 4 cells-12-02631-f004:**
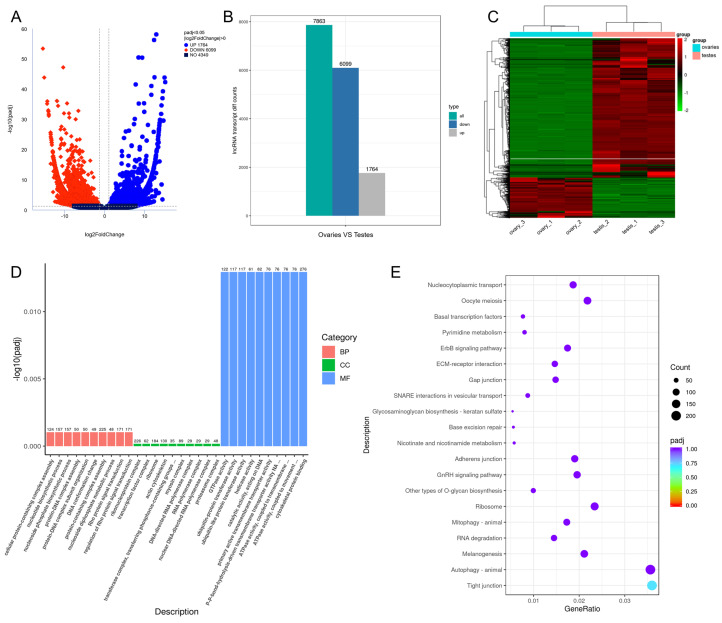
Analysis of DE lncRNAs between ovaries and testes. (**A**) Volcano map of DE lncRNAs. The blue dots represent the up-regulated lncRNAs, and red dots represent the down-regulated lncRNAs in ovaries compared to testes. (**B**) The number of DE lncRNAs. (**C**) Cluster analysis of DE lncRNAs. (**D**) The GO terms with the lowest *p*-value by DE lncRNA target genes from co-expression analysis. BP: biological process; CC: cell component; MF: molecular function. (**E**) The KEGG pathways with the lowest *p*-value by DE lncRNA target genes from co-expression analysis.

**Figure 5 cells-12-02631-f005:**
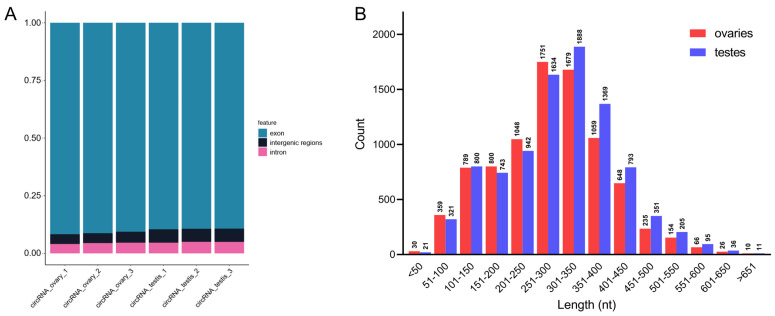
Identification of novel circRNAs. (**A**) Proportion of regions from which circRNAs were transcribed. (**B**) Length distribution of circRNAs.

**Figure 6 cells-12-02631-f006:**
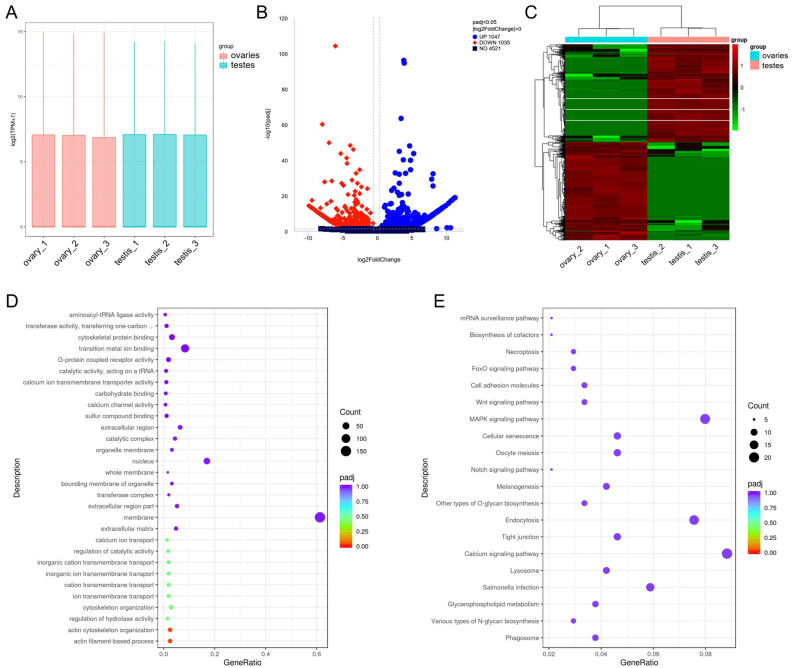
Expression and functional analysis of circRNAs in the gonads of common carp. (**A**) TPM box plot showing the expression profile of each sample. The horizontal axis is the value of log_10_ (TPM + 1). (**B**) The vocal map of DE circRNAs between ovaries and testes. Blue dots represent the up-regulated circRNAs, and red dots represent the down-regulated circRNAs in ovaries compared to testes. (**C**) Hierarchical clustering heatmap analysis of DE circRNAs between ovaries and testes. (**D**) The GO terms with the lowest *p*-value by genes transcribing DE circRNAs. (**E**) The KEGG pathways with the lowest *p*-value by genes transcribing DE circRNAs.

**Figure 7 cells-12-02631-f007:**
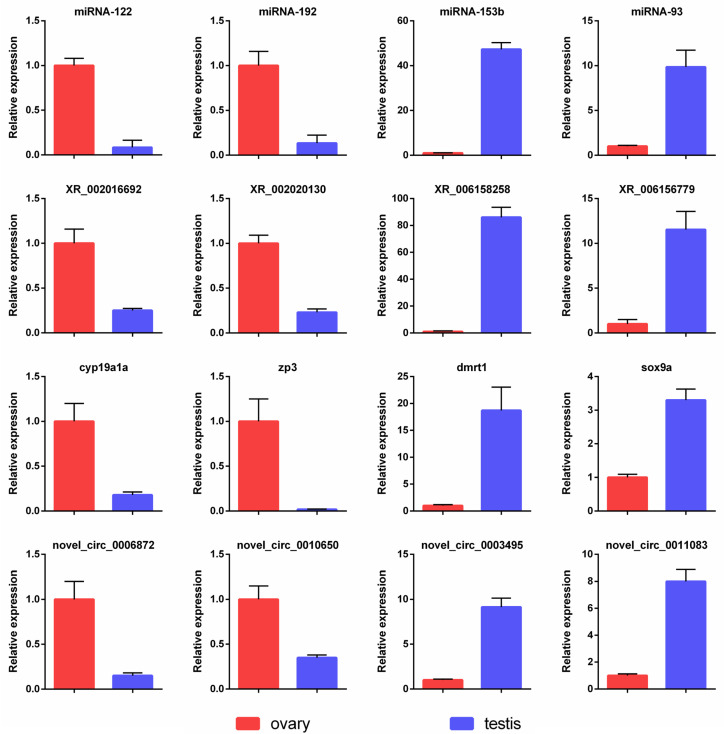
qPCR validation of miRNA, lncRNA, and circRNA sequencing data. The data shown are the mean ± SEM of three replicates, *p* ≤ 0.05.

**Figure 8 cells-12-02631-f008:**
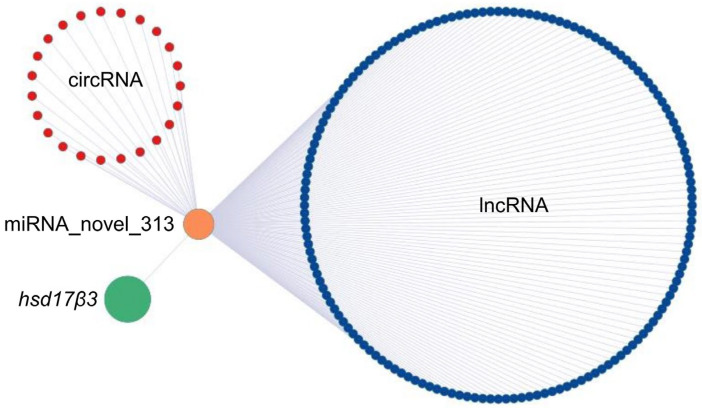
A case of ceRNA crosstalk by Cytoscape. Blue and red nodes represented lncRNAs and circRNAs, respectively, and the details of lncRNAs and circRNA are shown in Appendix A; the edge represents the interaction between the two nodes.

## Data Availability

The sequence data presented in the current study have been submitted to the National Center for Biotechnology Information (NCBI) BioProject database (Accession Numbers: PRJNA1020596).

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
