# Peer review of "Differential Expression of miRNAs, lncRNAs, and circRNAs between Ovaries and Testes in Common Carp (Cyprinus carpio)"

_cells, 2023, doi:10.3390/cells12222631_

Round 1

Reviewer 1 Report

Comments and Suggestions for Authors I read with pleasure and interest this article “Differential expression of miRNAs, lncRNAs, and circRNAs between ovaries and testes in common carp”. The aim was to investigate the expression profiles of ncRNAs, including miRNAs, lncRNAs, and circRNAs, in the gonads to comprehend the roles of ncRNAs in sex differentiation in common carp. So the study shows quite interesting results, but generally speaking, the most important results were that a substantial number of differentially expressed ncRNAs in ovaries and testes were identified, and they may intervene in the hormonal pathway and thus play a role in sex differentiation. However, the manuscript requires some information before considering publication.

After total RNA extraction by TRIzol, the RNA samples were not treated by DNAse, and the NanoDrop 260/280 and 260/230 ratios were not shown. So, it would be very interesting to show the Agilent Bioanalyzer electropherograms of RIN 7,5 and 8,5, to show that there was no DNA in the RNA samples. Moreover, it is known that contamination of samples with TRIzol is common and that this in turn influences cDNA synthesis, so the NanoDrop 260/280 and 260/230 ratios would give the purity information.

Reviewer 2 Report

Comments and Suggestions for Authors

Comments about the manuscript:

“Differential Expression of miRNAs, lncRNAs and circRNAs between Ovaries and Testes in Common Carp”

In carp, an economically important species,females grow faster than males. Noncoding RNAs (ncRNAs) exhibit versatile regulatory activities for several biological processes. However, their role in the sexual differentiation of common carp is little studied. The work presented here concerns the demonstration of the expression of several types of ncRNA (miRNA, lncRNA, circRNA) in male and female gonads, in order to understand their role in sexual differentiation in carp. The study shows, among other things, that certain miRNAs could intervene in hormonal synthesis and thus play a role in sexual relations. cRNA-mRNA interactions were also analyzed.

This substantial and complex work seems to me to be well done. The manuscript gives many results that require a lot of attention from the reader. However, the manuscript requires some improvements before considering its publication.

Title and text: add the Latin name (genus and species) of the common carp studied here.

Page 2, line 85. “Ovary and testis tissues were dissected”: The number of individuals studied must be specified: how many males? How many females?

Page 2, line 88. “following the manufacturer's instructions” is not sufficient for a scientific article: please briefly describe the method and technique.

Page 6, figure 2D “Volcano map of differentially expressed miRNAs”: Specify what the colors red or blue mean (ovaries, testicles, I think?)

“CC: cell component”: I'm not sure I understood the figure very well: there is no cellular component if I judge by the absence of green bars. Please check and explain if necessary.

Page 9, figure 4A “Volcano map of DE lncRNAs”: specify in the legend what the blue and red colors used indicate.

Page 9, figure 4D “co-expression analysis”: specify what BP, CC, MF mean (as in Figure 2).

Page 10, figure 6B: specify in the legend what the colors used indicate.

Reviewer 3 Report

Comments and Suggestions for Authors

The authors Mingxi Hou et al. have submitted a manuscript in cells (Manuscript No. cells-2682014), entitled “Differential Expression of miRNAs, lncRNAs and circRNAs between Ovaries and Testes in Common Carp”.

The manuscript reports a study conducted on common carp, a traditional fish in China, the aim is to investigate the expression of genes for non-coding RNA and mRNA in carp gonads (ovaries and testes), these transcripts includes small and long non coding RNA and coding mRNA (miRNA, lncRNA, circRNA and mRNA) and are involved in many biological roles. In this manuscript, the expression of these genes in the ovaries and testis of carp were analyzed and compared.

In fact, in the present  study of the transcriptoma from carp ovaries and testes were cloned as cDNA in different library and sequenced. The authors, report an expression patterns and some of these transcripts were certified by quantitative PCR analysis and bioinformatics analyses.

The results show the expressed genes for all type of RNAs analyzed, showing specific and differential expression in ovaries and testes and identifieng a pool of RNAs associated with sex differentiation. In addition, the functionally interaction between the different type of RNAs are studied, acts to regulate gene expression by preventing them from binding to their target mRNAs regulating the activites.

I believe that all the data of this study could be useful to increase the knowledge on the novel RNAs and show a major complessity of trascriptomic until now konow.

Finally, I consider the document in this form suitable for publication in cells.

Comments on the Quality of English Language

The authors Mingxi Hou et al. have submitted a manuscript in cells (Manuscript No. cells-2682014), entitled “Differential Expression of miRNAs, lncRNAs and circRNAs between Ovaries and Testes in Common Carp”.

The manuscript reports a study conducted on common carp, a traditional fish in China, the aim is to investigate the expression of genes for non-coding RNA and mRNA in carp gonads (ovaries and testes), these transcripts includes small and long non coding RNA and coding mRNA (miRNA, lncRNA, circRNA and mRNA) and are involved in many biological roles. In this manuscript, the expression of these genes in the ovaries and testis of carp were analyzed and compared.

In fact, in the present  study of the transcriptoma from carp ovaries and testes were cloned as cDNA in different library and sequenced. The authors, report an expression patterns and some of these transcripts were certified by quantitative PCR analysis and bioinformatics analyses.

The results show the expressed genes for all type of RNAs analyzed, showing specific and differential expression in ovaries and testes and identifieng a pool of RNAs associated with sex differentiation. In addition, the functionally interaction between the different type of RNAs are studied, acts to regulate gene expression by preventing them from binding to their target mRNAs regulating the activites.

I believe that all the data of this study could be useful to increase the knowledge on the novel RNAs and show a major complessity of trascriptomic until now konow.

Finally, I consider the document in this form suitable for publication in cells.
